# Publication rates of research projects of an internal funding program of a university medical center in Germany: A retrospective study (2004–2013)

**Susanne Deutsch**[1], **Silke Reuter**[2], **Astrid Rose**[2], **René Tolba**[1]*

**1** Faculty of Medicine, Institute for Laboratory Animal Science & Experimental Surgery, RWTH Aachen University, Aachen, Germany, **2** Faculty of Medicine, Dean's Office, RWTH Aachen University, Aachen, Germany

* rtolba@ukaachen.de

## Abstract

### Objectives

Non-publication and publication bias are topics of considerable importance to the scientific community. These issues may limit progress toward the 3R principle for animal research, promote waste of public resources, and generate biased interpretations of clinical outcomes. To investigate current publishing practices and to gain some understanding of the extent to which research results are reported, we examined publication rates of research projects that were approved within an internal funding program of the Faculty of Medicine at a university medical center in Germany, which is exemplary for comparable research funding programs for the promotion of young researchers in Germany and Europe.

### Methods

We analyzed the complete set (n = 363) of research projects that were supported by an internal funding program between 2004 and 2013. We divided the projects into four different proposal types that included those that required an ethics vote, those that included an animal proposal, those that included both requirements, and those that included neither requirement.

### Results

We found that 65% of the internally funded research projects resulted in at least one peer-reviewed publication; this increased to 73% if other research contributions were considered, including abstracts, book and congress contributions, scientific posters, and presentations. There were no significant differences with respect to publication rates based on (a) the clinic/institute of the applicant, (b) project duration, (c) scope of funding or (d) proposal type.

### Conclusion

To the best of our knowledge, this is the first study to explore publication rates associated with early-career medical research funding. As >70% of the projects ultimately generated

**Data Availability Statement:** All relevant data are within the manuscript and its Supporting Information files.

**Funding:** This work is associated with the research project "EMBARC" funded by the German Federal Ministry of Education and Research (BMBF 031L0131B, https://www.bmbf.de/). SD was funded in part by the German Federal Ministry of Education and Research (BMBF 031L0131B, https://www.bmbf.de/). The funder had no role in study design, data collection and analysis, decision to publish, or preparation of the manuscript.

**Competing interests:** The authors have declared that no competing interests exist.

some form of publication, the program was overall effective toward this goal; however, non-publication of research results is still prevalent. Further research will explore the reasons underlying non-publication. We hope to use these findings to develop strategies that encourage publication of research results.

## Introduction

Non-publication and publication bias with respect to results dissemination are topics of substantial concern in the scientific community [1]. Among the problems associated with these phenomena, non-publication of results may lead to the unnecessary repetition of animal experiments. In the absence of a timely publication of research results, poorly planned or even failed animal experiments may be repeated by other researchers; this clearly contradicts the 3R principle which is directed toward minimizing the number of animals used in experiments [2,3]. Equally troubling is the non-publication of clinical trials. A recently published study revealed that only ~7% of the results from all clinical trials that are carried out at university medical centers (UMCs) in Germany are published in the European Union clinical trials registry [4]. Although this percentage does not reflect the extent to which clinical trials are published in the scientific literature, this fact remains disconcerting and a topic of substantial concern. Underreporting of clinical trials has substantial negative consequences that include the potential for direct harm to patients, delayed medical progress and waste of public funds. Non-publication and underreporting of results both violate the ethical principles of the Declaration of Helsinki that provides ethical guidance for all medical research involving human subjects as per the World Medical Association (WMA) [5–7]. Several studies have been published that have evaluated publication rates of both animal and clinical research [8–14]. However, we do not have sufficient information on publication rates associated with early-stage projects; these projects by nature have a comparatively high potential for failure and as such results may not be reported or published.

This study is designed to address the issue of publication rates of research projects from an internal funding program for early-career scientists based at a German UMC (https://bit.ly/2BlAaTj). The internal funding program described is exemplary for comparable funding programs for the promotion of young researchers in Germany and Europe. This type of funding is available at Faculties of Medicine in Germany, like the Fortüne program of the Faculty of Medicine of the Eberhard Karls University Tübingen (https://bit.ly/3eB9vjX) or the Fortune *program* of the Faculty of Medicine of the University of Cologne (https://bit.ly/2Xh73Jp), and in Europe, like the Kootstra Talent Fellowship program of Maastricht University (https://bit.ly/2McNFH9) or Pump-Priming programs of the University of Cambridge (https://bit.ly/36NFg6s). The study presents the evaluation of an internal research funding program in terms of its quality characteristics, which are represented by key figures, namely the ratio of published and non-published research projects. The present study provides information on the true extent of publication rates of research projects of an internal research funding program that is exemplary for comparable funding programs in Germany and Europe, and thus gives an impression of the dissemination of research results and the visibility and sustainability of internal research funding [15].

For this study, we investigated all research proposals that were funded between the years 2004 and 2013, including those that featured animal and/or clinical research. This specific funding program was established in the mid-1990s as part of a research innovation program

from what was previously known as Ministry of Science and Research of the State of North Rhine-Westphalia (NRW). The primary objective of this program was to provide start-up funding that would ultimately facilitate acquisition of third-party funds from major research funding organizations, including the German Research Foundation (Deutsche Forschungsgemeinschaft; DFG) and/or the Federal Ministry of Education and Research (Bundesministerium für Bildung und Forschung; BMBF). Publication of results of earlier research projects in high-profile journals is a prerequisite for consideration by either of these major funding bodies. Internal funding programs provide young scientists with the critical opportunity to fulfil these requirements for the successful acquisition of third-party funding. Physicians with a doctoral degree and other scientists with equivalent scientific credentials who were employed at the Faculty of Medicine and who had not yet habilitated (i.e., qualified as a postdoctoral lecturer at a German university) and/or who will not reach the junior or associate professor level at the time of an interim evaluation are eligible for early-career funding. At present, the type and scope of projects that can be considered for funding range from small to more complete, or full applications. Small applications are particularly suited to exploratory research projects where the success of the project is subject to considerable risk. As of this writing, the upper limit of funding for a small application is 25,000 Euros; full applications can be funded up to a limit of 105,000 Euros. Although there were adjustments to the funding limits during the earlier years (2004–2013), the relative funding levels throughout were similar to those included here. Funding can be used for all types of projects including (but not limited to) those featuring patients, human biomaterials, and animals. All applications were reviewed by at least two or three competent scientists of the Faculty of Medicine at the university. An evaluation sheet was used that featured systematic predefined rating scales and an open evaluation text field. Based on the points awarded, a ranking list was created, on the basis of which as many projects were approved as there were free funds in the budget of the funding program in the respective year.

The aim of this study was to determine the publication rates of the projects approved within the framework of this funding program in order to draw conclusions about possible publication predictors. We were specifically interested in whether non-publication was associated with a specific type of project, notably those that required an ethics vote and/or an application for permission to perform animal experiments (animal proposal).

## Material and methods

This study was carried out in 2020 in cooperation with the Vice Dean for Research and Junior Researchers and individuals responsible for organizing the internal funding program at the Faculty of Medicine. For this purpose, internal information was prepared by the Office of the Vice Dean and independently evaluated by authors SD and RT. The study represents primary research as the authors have collected data from the internal research funding program for research purposes for the first time. Used and presented methodology of this study has already been applied in a different paper recently published in PLOS ONE by authors who also conducted this study [1].

### Data collection

All research proposals funded between 2004 and 2013 were included in this study. As this study focuses on publication rates, this period was chosen to ensure that researchers responsible for even the most recent projects approved in 2013 had sufficient time to publish results from the project after completion. On receipt of research funding, all applicants are committed to perform a full evaluation after completion of the project and are required to provide

information on all project-related publications, including peer-reviewed articles, published abstracts, book and congress contributions as well as relevant oral and poster presentations. The dataset used to analyze the publication rates contained the following information: (a) year of approval, (b) applicant with associated clinic/institute, (c) title of the research project, (d) approved project duration, (e) scope of funding, (f) information on whether an ethics vote and/or an application for permission to perform animal experiments were required, and (g) details of all project-related publications, which distinguished between peer-reviewed articles and other types of result publications.

## Data preparation

In order to check whether or not all reported publications were subject to a peer-review process, corresponding journals were searched within the Web of Science and editorial guidelines for the peer-review process were checked. Additionally, the corresponding two-year impact factors (IFs) associated with the date of publication were retrieved from journal citation reports within the Web of Science. For research projects with no associated peer-reviewed publications, PubMed, Web of Science, Google Scholar and the university library were searched by author SD using the name of the applicant, the name of the institution and keywords from the title as search variables as previously described by Wieschowski et al. [1]; the search in this cited study was performed by author SD and showed high interrater reliability. All research projects were categorized as belonging to one of four groups: (1) neither ethics vote nor animal proposal was required, (2) ethics vote was required, (3) animal proposal was required or (4) both animal proposal and ethics vote were required. Applicants and corresponding research projects were also classified in two groups based on the applicants' institutional affiliations, including (a) clinics and institutes in which the researcher was required to take part in direct patient care and (b) institutes in which researchers had no patient care responsibilities.

## Registration

The study protocol was preregistered at Open Science Framework (https://osf.io/hdc47/).

## Results

### Demographic data

A total of 363 research projects that were funded between 2004 and 2013 were included in this study. Of these 363 studies, 104 (29%) required an ethics vote only, 150 (41%) required permission to perform animal experiments, 17 (5%) required both ethics vote and permission to perform animal experiments, and 92 (25%) required neither. Taken together, 167 (46%) of the funded projects required permission to perform animal experiments. Fig 1 includes the percentages of each proposal type as indicated on an annual basis. On average 36.3 ± 6.72 research projects were funded each year; the range per year included a minimum of 29 (in 2007) and a maximum of 52 (in 2013) funded research projects. Of the 70 clinics and institutes associated with the UMC, there are currently 43 (61%) in which participants are actively involved in patient care and 27 (39%) in which participants do not provide patient care. Within the framework of the funding program, a total of 261 (72%) research projects were submitted and approved by applicants from clinics and institutes in which researchers were required to take part in direct patient care and 102 (28%) from institutes where researchers had no patient care responsibilities.

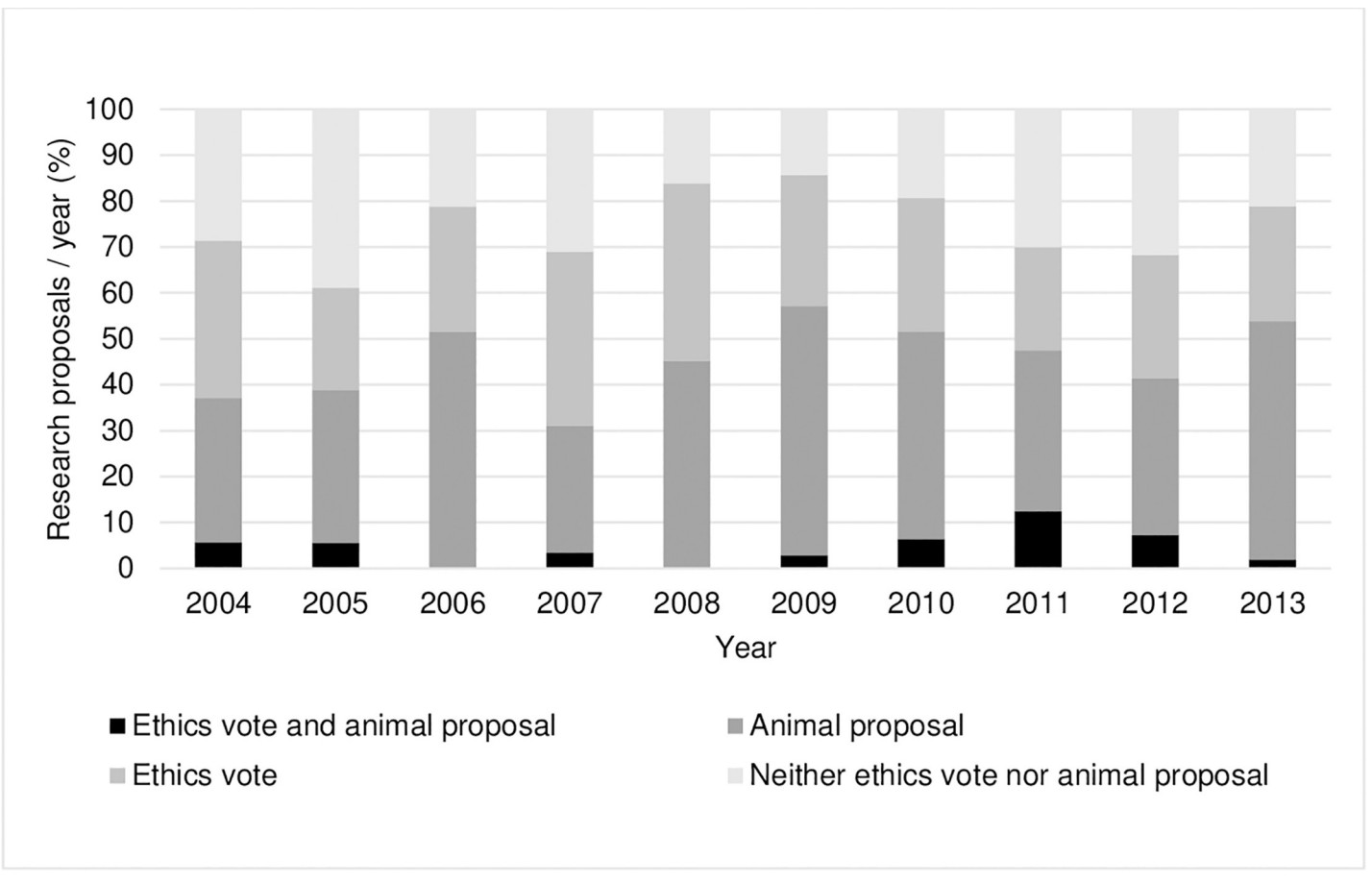

**Fig 1. Percentages of approved research proposals that required an ethics vote and/or an animal proposal or neither ethics vote nor animal proposal for each year within the ten-year period.**

### Publication rates

Over the ten-year period, 65% of the approved research proposals produced at least one peer-reviewed publication. If one extends the concept of publication rate to include published abstracts, book and congress contributions, scientific posters, and presentations, 73% of the approved proposals ultimately generated research publications. We observed no specific trends suggesting either an increase or decrease in publication rates over time; rather the years 2008 and 2012 could be identified as years with particularly high publication rates (Table 1). For research projects that are associated with at least one project-related peer-reviewed publication, the number of publications per project ranged from a minimum of 1 to maximum of 22, with a mean value of 2.63 ± 2.35 publications per project.

Regarding the impact factors (IFs) of all peer-reviewed publications, a mean value of 4.676 ± 3.994 and a median of 3.800 (Minimum = 0.085, Maximum = 59.558) could be identified. The sum of the IFs of all peer-reviewed publications per individual project ranged from a minimum of 0.085 to a maximum of 116.861, with a median of 7.024 and a mean value of 11.942 ± 13.404. In total, there were 31 peer-reviewed publications published in journals for which no IF was available at the time of publication; these publications were not taken into account in our calculations relating to the IF. There were no significant differences in publication rates based on $X^2$ tests with regard to (a) the clinic/institute of the applicant (with or

**Table 1. Publication rates.**

| | Total number of approved research proposals | Percentage of approved research proposals that generated at least one peer-reviewed publication | Percentage of approved research proposals that generated peer-reviewed or other forms of result publication[1] |
|---|---|---|---|
| **Year of approval of the research proposal** | | | |
| 2004 | 35 | 57% | 74% |
| 2005 | 36 | 69% | 83% |
| 2006 | 33 | 61% | 82% |
| 2007 | 29 | 59% | 66% |
| 2008 | 31 | 84% | 84% |
| 2009 | 35 | 71% | 71% |
| 2010 | 31 | 65% | 71% |
| 2011 | 40 | 58% | 63% |
| 2012 | 41 | 80% | 85% |
| 2013 | 52 | 50% | 60% |
| *Total* | *363* | *65%* | *73%* |
| **Proposal type[3]** | | | |
| Neither ethics vote nor animal proposal required | 92 | 64% | 75% |
| Ethics vote required | 104 | 66% | 77% |
| Animal proposal required | 150 | 64% | 70% |
| Ethics vote and animal proposal required | 17 | 65% | 71% |
| **Scope of funding[3]** | | | |
| Small application ($\leq$ 25 k€) | 81 | 59% | 67% |
| Full application (25–105 k €) | 282 | 66% | 75% |
| **Project duration (months)[3]** | | | |
| < 12 | 4 | 75% | 100% |
| 12 | 130 | 62% | 75% |
| 18 | 23 | 65% | 74% |
| 24 | 206 | 67% | 72% |
| **Clinic/Institute[2,3]** | | | |
| Without tasks related to patient care | 102 | 70% | 78% |
| With tasks related to patient care | 261 | 63% | 71% |

[1]This classification includes peer-reviewed publications as well as published abstracts, book and congress contributions, scientific posters, and presentations.

[2]This classification refers to the affiliation of the applicant.

[3]$X^2$ tests were performed with no significance regarding proposal type, scope of funding, project duration and clinic/institute of the applicant.

without patient care responsibilities), (b) project duration, (c) scope of funding and (d) proposal type as defined above.

## Discussion

To the best of our knowledge, this is the first study that provides insights into demographics and publication rates associated with an early-career internal funding program of a major German UMC, which is exemplary for comparable research funding programs for the promotion

of young researchers in Germany and Europe. We found that 65% of the research projects generated at least one peer-reviewed result publication, and 73% generated some form of result publication, including published abstracts, book and congress contributions, scientific posters, and presentations.

## Key findings

The percentage of approved research proposals with various types of result publication was over 70% in almost every category (year of approval, proposal type, scope of funding, project duration, clinic/institute of the applicant); thus a full two-thirds of all funded projects have presented their results in one or more formal ways to the scientific community. We identified no factors that influenced the publication rates or number of publications per research project, although we considered (a) the clinic/institute of the applicant (with or without patient care responsibilities), (b) project duration, (c) scope of funding and (d) proposal type (i.e., those requiring an ethics vote and/or animal proposal *vs.* those that did not).

## Publication rate

To generate a clear understanding regarding the relevance of the results found, it is essential to relate our findings to those of comparable studies in the literature. Several studies have evaluated publication rates with a focus on abstracts presented at scientific conferences [9,11,12,16–20]. For example, Meral et al. [11] reported that 40.9% of all presented abstracts, podium discussions and poster presentations at the annual congress of the European Society for Surgical Research between 2008 and 2011 were ultimately published in peer-reviewed format. Narain et al. [16] and Mullen et al. [17] found comparable overall publication rates of 43.8% and 42.9%, respectively for similar specific annual congresses. Crawford et al. [18] reported that (44%) of the oral abstract presentations from scientific congresses were published as peer-reviewed contributions; likewise, Bonfield et al. [19] reported publication rates of 60.6% and 40.6% for two independent scientific meetings. These publications did not differentiate among types of studies, and as such, it is not clear whether the results included presentations based on animal research. There is currently very little information available on the publication rates that focus on studies with animal research. For example, a follow-up of animal research-based studies reported among the research abstracts presented at the 2008 Society of Critical Care Medicine Conference revealed a publication rate of 62% [9], which is comparable to the publication rate identified here in our study. Nevertheless, the low publication rates documented in the earlier studies discussed above (mostly < 50%) suggest that there is an urgent need for improvement. This point of view is supported by data from the systematic reviews of Scherer et al. [12,20], who found an overall publication rate of conference abstracts across all disciplines to be only 44.5% (95% confidence interval (CI), 43.9% to 45.1%; 2007 findings); the rate dropped to 37.3% (95% CI, 35.3% to 39.3%) in 2018. When comparing these results to those revealed by our study, it should be noted that all studies focused on individual conferences and as such, the abstracts included were those in a specific research field or discipline; it is not certain that the results obtained are directly relevant to other conferences or disciplines. By contrast, our study covers all research conducted at one UMC. In principle, one might assume that the publication rate determined on the basis of conference abstracts would be higher than those that emerged from our study. The research presented at a conference has already been vetted to some extent and has been prepared for presentation to an audience of one's peers. As such, data from the literature suggest that the publication rate determined for the single German UMC funding program is comparatively high and reflects strong achievement throughout.

There are several published studies that have focused on publication rates of research proposals that required ethical approval [10,21,22]. The systematic review of Schmucker et al. [10] indicated that 26% to 76% of the studies approved by research ethics committees were associated with peer-reviewed publications, with a pooled estimate publication rate of 46% (95% Cl, 40.2% to 52.4%). Similarly, the publication rates of studies included in trial registries ranged from 23% to 76% with a pooled estimate of 54.2% (95% Cl, 42% to 65.9%). Although this study shows a great heterogeneity of the studies with respect to the countries in which the studies were conducted, the nature of the study sample and the methodology chosen to identify corresponding journal publications, these results were overall in alignment with those recently published by Wieschowski et al. [21], in which all registered clinical studies completed between 2009 and 2013 for all German UMCs were tracked. This study identified a publication rate of 39% at two years and a 74% publication rate at six years after study completion. Focusing on clinical research after ethical approval at one large German UMC, Blümle et al. [22] found that 52% of all approved as well as conducted clinical studies had generated published results. However, it is critical to recognize that all these studies illustrate the ongoing and widespread underreporting of clinical research, which contradicts the guidance provided in a recent publication by the World Health Organization (WHO) [23]. In our study, which focuses on the publications from early-career scientists supported by an internal funding program, a similar, if not higher rate was observed. Among the reasons for the high publication rate, this may relate directly to the nature and the conception of the funding program. Researchers are aware of the fact that they must ultimately report their results in a formal, peer-reviewed format in order to have the opportunity to apply for a second round of grant funding. The described German UMC funding program is aimed at highly motivated young researchers whose further scientific careers are highly dependent on their publication performance and who therefore have a strong impetus to publish their results. Another possible explanation for the comparatively high publication rate associated with our study is the fact that we included a prolonged period between time of initial funding to publication date; this was not considered in all studies. In the case of our study, some of the intervals were quite long, as the final search for publications was performed at the beginning of 2020.

The publication rates found in our study were similar to those reported by Wieschowski et al. [1], who investigated the publication rates of studies that included animal research at two German UMCs over a comparable period of time; a publication rate of 67% was determined in this latter study. Although not only animal experiments were subject of this study, the comparison suggests that our results are both credible and within an already established range and scope.

Despite the comparatively high rates of research publication observed here and in the previous study [1], non-publication remains prevalent. In previous studies that directly addressed these issues, most researchers cite lack of time and the perception that their research is low priority and not deemed valuable, notably among studies with primarily negative results [24,25]. Other issues include technical problems and difficulties with co-authors, supervisors and/or peer reviewers [8]. To deepen the understanding of the difficulties associated with the publication process, it might be useful to conduct a formal survey of the young researchers funded by this program in order to identify specific reasons for non-publication. Results of this type of study may yield concrete improvements, incentives, and supportive measures that would be directed toward closing the publication gap and thus increasing the publication rate.

## Limitations

We are aware of several limitations of this study. First, this study provides insights into the internal funding program of one German Faculty of Medicine only; we cannot assume that

our findings will have immediate applicability to other scientific areas of the university or to similar programs at other research institutions. Second, the fact that all research projects were funded within a unique internal funding program limits our ability to identify the full range of factors that might influence the rate or number of research publications. The funding structure required researchers to present their findings in publication form regardless of factors that include the specific clinic/institute of the applicant, duration of the funded project, scope of funding or proposal type. Reporting of results is mandatory and essential to maintain the opportunity, which is to permit the application to obtain either a second approved application within the internal funding program or external third-party research support. Third, we note a comparatively low publication rate of (50% for peer-reviewed, 60% for all publications) for the final year of our study (2013). This may suggest that not all results have been published at this date in time. We recognize that the funding program permits cost-neutral project extensions; as such, projects that were approved in 2013 with a maximum duration of two years may not have been completed in 2015 but in one or more of the years to follow. This observation suggests the likelihood that there will be additional publications forthcoming from this final group of grantees. Fourth, the papers reported by the applicants were not cross-checked for their primary affiliation; as such, we cannot rule out the possibility that there are some reported publications that may not have been funded directly by these grants. Finally, it remains possible that results from some research projects were published but not reported by the applicant and also not detected by the follow-up search performed in this study. In order to track more recent publications, the university established an internal publication database; this will facilitate identification of all publications published by faculty at the university in future, since there is an specific and ongoing obligation to report all publications with university affiliation.

## Conclusions

To the best of our knowledge, this the first study to explore factors that could influence publication rates of projects supported by a medical research funding program that aims to support early-career scientists in their efforts to apply for third-party funding from well-established major research funding organizations. Since the funding program described is comparable to similar funding programs in Germany and Europe, the findings made are important and beneficial for the design of a funding landscape for the promotion of young scientists. Our results indicate a relatively high publication rate among the projects funded by this program. As such, we conclude that this program is effective at fostering a culture that promotes publication in order to enhance visibility of research results. Nevertheless, further investigation needs to be conducted to determine whether publication of negative results is also carried out within the context of this program and/or whether this specific aspect needs to be improved.

## Supporting information

**S1 Table. Raw data associated with each of the research projects included (2004–2013); data that might permit identification of persons involved have been redacted.** (XLS)

## Acknowledgments

The authors thank Susanne Wieschowski and Daniel Strech for their feedback during the preparation of the manuscript.

## Author Contributions

**Conceptualization:** Susanne Deutsch, René Tolba.

**Data curation:** Susanne Deutsch, Silke Reuter, Astrid Rose.

**Formal analysis:** Susanne Deutsch, René Tolba.

**Funding acquisition:** René Tolba.

**Investigation:** Susanne Deutsch.

**Methodology:** Susanne Deutsch, René Tolba.

**Supervision:** Silke Reuter, Astrid Rose, René Tolba.

**Writing – original draft:** Susanne Deutsch.

**Writing – review & editing:** Susanne Deutsch, Silke Reuter, Astrid Rose, René Tolba.

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
