## [Decision Letter · Decision Letter 0]

7 Oct 2020

PONE-D-20-14186

Publication Rates of Research Projects of an Internal Funding Program of a University Medical Center in Germany: A Retrospective Study (2004–2013)

PLOS ONE

Dear Dr. Deutsch,

Thank you for submitting your manuscript to PLOS ONE. After careful consideration, we feel that it has merit but does not fully meet PLOS ONE’s publication criteria as it currently stands. Therefore, we invite you to submit a revised version of the manuscript that addresses the points raised during the review process.

Below, please find the comments and revisions required by the academic reviewer who assessed your manuscript. You will find that most of them are really minor. So, please carefully address all them (including the response to each comment or query at the rebuttal letter) and resubmit it for a further evaluation.

We look forward to receiving your revised manuscript.

Kind regards,

Sergio A. Useche, Ph.D.

Academic Editor

PLOS ONE

Journal Requirements:

Reviewers' comments:

Reviewer's Responses to Questions

**Comments to the Author**

1. Is the manuscript technically sound, and do the data support the conclusions?

Reviewer #1: Yes

2. Has the statistical analysis been performed appropriately and rigorously? 

Reviewer #1: Yes

3. Have the authors made all data underlying the findings in their manuscript fully available?

Reviewer #1: Yes

4. Is the manuscript presented in an intelligible fashion and written in standard English?

Reviewer #1: Yes

5. Review Comments to the Author

Reviewer #1: I have a few minor suggestions.

1. I suggest adding a footnote to Table 1 indicating that Chi square tests were performed with no significance. If tests for trend were also done I would include a footnote referencing that test as well.

2. The sentence starting on line 275 "When questioned..."; I would make it clear that this information comes from previous research (not the current study) and move at least one of the citations: 8,24,25 up to support this statement.

3. It seems like the first part of reference 8 may have been deleted, please review.

6. PLOS authors have the option to publish the peer review history of their article (what does this mean?). If published, this will include your full peer review and any attached files.

Reviewer #1: No

---

## [Author Response · Author response to Decision Letter 0]

9 Oct 2020

Reviewer #1: 1. I suggest adding a footnote to Table 1 indicating that Chi square tests were performed with no significance. If tests for trend were also done I would include a footnote referencing that test as well.

Authors’ response: We thank the reviewer for the suggestion and added the following footnote to Table 1. No tests for trends were carried out. 

3 Χ2 tests were performed with no significance regarding proposal type, scope of funding, project duration and clinic/institute of the applicant.

Original manuscript: - 

Revised manuscript with track changes: Line 180 - 181

Reviewer #1: 2. The sentence starting on line 275 "When questioned..."; I would make it clear that this information comes from previous research (not the current study) and move at least one of the citations: 8,24,25 up to support this statement.

Authors’ response: Thank you for this valuable comment. We have revised these sentences.

In previous studies that directly addressed these issues, most researchers cite lack of time and the perception that their research is low priority and not deemed valuable, notably among studies with primarily negative results [24, 25]. Other issues include technical problems and difficulties with co-authors, supervisors and/or peer reviewers [8].

Original manuscript: Line 275 - 278

Revised manuscript with track changes: Line 276 - 280

Reviewer #1: 3. It seems like the first part of reference 8 may have been deleted, please review.

Authors’ response: We thank the reviewer for carefully reading the manuscript. However, reference 8 is correctly indicated. It is a reference of Gerben ter Riet, a researcher from the Netherlands.

Original manuscript: Line 343

Revised manuscript with track changes: Line 345

---

## [Decision Letter · Decision Letter 1]

16 Nov 2020

Publication Rates of Research Projects of an Internal Funding Program of a University Medical Center in Germany: A Retrospective Study (2004–2013)

PONE-D-20-14186R1

Dear Dr. Tolba,

We’re pleased to inform you that your manuscript has been judged scientifically suitable for publication and will be formally accepted for publication once it meets all outstanding technical requirements.

Kind regards,

Sergio A. Useche, Ph.D.

Academic Editor

PLOS ONE

Additional Editor Comments (optional):

Reviewers' comments:

Reviewer's Responses to Questions

**Comments to the Author**

1. If the authors have adequately addressed your comments raised in a previous round of review and you feel that this manuscript is now acceptable for publication, you may indicate that here to bypass the “Comments to the Author” section, enter your conflict of interest statement in the “Confidential to Editor” section, and submit your "Accept" recommendation.

Reviewer #1: All comments have been addressed

2. Is the manuscript technically sound, and do the data support the conclusions?

Reviewer #1: Yes

3. Has the statistical analysis been performed appropriately and rigorously? 

Reviewer #1: Yes

4. Have the authors made all data underlying the findings in their manuscript fully available?

Reviewer #1: Yes

5. Is the manuscript presented in an intelligible fashion and written in standard English?

Reviewer #1: Yes

6. Review Comments to the Author

Reviewer #1: (No Response)

7. PLOS authors have the option to publish the peer review history of their article (what does this mean?). If published, this will include your full peer review and any attached files.

Reviewer #1: No

---

## [Editor Report · Acceptance letter]

18 Nov 2020

PONE-D-20-14186R1 

Publication Rates of Research Projects of an Internal Funding Program of a University Medical Center in Germany: A Retrospective Study (2004–2013) 

Dear Dr. Tolba:

I'm pleased to inform you that your manuscript has been deemed suitable for publication in PLOS ONE. Congratulations! Your manuscript is now with our production department. 

Kind regards, 

on behalf of

Dr. Sergio A. Useche 

Academic Editor

PLOS ONE